# The Role of Endothelial Dysfunction in Peripheral Blood Nerve Barrier: Molecular Mechanisms and Pathophysiological Implications

**DOI:** 10.3390/ijms20123022

**Published:** 2019-06-20

**Authors:** Jessica Maiuolo, Micaela Gliozzi, Vincenzo Musolino, Cristina Carresi, Saverio Nucera, Roberta Macrì, Miriam Scicchitano, Francesca Bosco, Federica Scarano, Stefano Ruga, Maria Caterina Zito, Francesca Oppedisano, Rocco Mollace, Sara Paone, Ernesto Palma, Carolina Muscoli, Vincenzo Mollace

**Affiliations:** 1Interregional Research Center for Food Safety and Health (IRC-FSH), Department of Health Sciences, University “Magna Græcia” of Catanzaro, Campus Universitario di Germaneto, 88100 Catanzaro, Italy; jessicamaiuolo@virgilio.it (J.M.); micaela.gliozzi@gmail.com (M.G.); xabaras3@hotmail.com (V.M.); carresi@unicz.it (C.C.); saverio.nucera@hotmail.it (S.N.); robertamacri85@gmail.com (R.M.); miriam.scicchitano@hotmail.it (M.S.); francescabosco@libero.it (F.B.); federicascar87@gmail.com (F.S.); rugast1@gmail.com (S.R.); mariacaterina.zito@libero.it (M.C.Z.); oppedisanof@libero.it (F.O.); mollace.rocco@gmail.com (R.M.); sara.paone06@gmail.com (S.P.); palma@unicz.it (E.P.); muscoli@unicz.it (C.M.); 2Nutramed Societa’ Consortile A Responsabilita’ Limitata (S.c.a.r.l.), Complesso Ninì Barbieri, Roccelletta di Borgia, 88021 Catanzaro, Italy; 3Istituto di Ricovero e Cura a Carattere Scientifico (IRCCS) San Raffaele, Via di Valcannuta 247, 00133 Rome, Italy

**Keywords:** Blood Nerve Barrier (BNB), nitric oxide, nitric oxide synthase (NOS), endothelial dysfunction, diabetic neuropathy, neuropathic pain, peripheral nerve injury, erectile dysfunction

## Abstract

The exchange of solutes between the blood and the nerve tissue is mediated by specific and high selective barriers in order to ensure the integrity of the different compartments of the nervous system. At peripheral level, this function is maintained by the Blood Nerve Barrier (BNB) that, in the presence, of specific stressor stimuli can be damaged causing the onset of neurodegenerative processes. An essential component of BNB is represented by the endothelial cells surrounding the sub-structures of peripheral nerves and increasing evidence suggests that endothelial dysfunction can be considered a leading cause of the nerve degeneration. The purpose of this review is to highlight the main mechanisms involved in the impairment of endothelial cells in specific diseases associated with peripheral nerve damage, such as diabetic neuropathy, erectile dysfunction and inflammation of the sciatic nerve.

## 1. Introduction

The correct functioning of the exchanges between blood and peripheral tissues oriented to the maintenance of tissue homeostasis, is guaranteed by the existence of systems of barriers which selectively regulate the communications between blood and tissues in order to act as a filter that ensures a sustainable level of tissue spread of substrates and soluble molecules. This system of barriers is particularly important both in the Central Nervous System (CNS) and in the peripheral nerve. However, although the blood-brain barrier (BBB) has been extensively studied in its implications in the development of neurodegenerative diseases [1], the mechanisms that regulate exchanges between blood and peripheral nerve, the so-called Blood Nerve Barrier (BNB), has been less studied, especially in its implications with peripheral neuropathies.

The barriers are mainly constituted by endothelial cells that are held together by very tight junction systems; thus affording a selective transport system able to guarantee the passage of the substances necessary for the maintenance of tissue homeostasis. At the CNS level, the function of the BBB is to ensure the perfect balance between extracellular brain fluids and plasma [2], while in the peripheral nervous system (PNS), this process is controlled directly by the BNB, which limits the entry of substances into the blood and maintains the homeostasis of the nervous tissue adjacent to the endothelium [3].

In the past, the endothelium was considered as a simple single-cell interface of cells located between the blood and the blood vessel wall of body tissues. At the end of the 1980s, however, it was discovered that endothelial cells are actively involved in the biochemical regulation of different vascular functions, so that the endothelium began to be considered as a true dynamic organ, with autocrine, paracrine and endocrine functions, as well as a semipermeable physical barrier that responds to chemical or mechanical stimuli generated by the two compartments that surround it [4]. Structurally, the endothelium of an adult man is made up of 10^13^/10^14^ cells capable of covering a surface of 1–7 m^2^ and it have a weight of about 1 kg [5]. Furthermore, endothelial cells guarantee, with their activity of secretion of chemical mediators, the local regulation of arterial flow, the fluidity of blood, and the passage of nutrients, hormones and macromolecules that modulate and support the surrounding tissues [6].

In particular endothelial cells, at the barrier level, modulate the district flows regulating not only the blood supply but also the series of bi-directional exchanges between plasma and tissue fluids [6] avoiding the overexposure of organs and tissues, such as nervous tissue, to the potentially toxic action of exogenous agents. This is also confirmed by the early consequences of the destruction of BBB endothelial cells in triggering neurodegenerative processes at central level [7]. Similarly, among other implications, alterations in endothelial function at the BNB level seem to be able to consolidate the development and the evolution of pathologies of the PNS, especially at the level of autonomic control of some visceral functions.

The present review is mainly aimed to verify the endothelial function in the regulation of BNB and its implications in the development of diseases that involve autonomic neurovascular regulation.

## 2. Restrictive Properties of the Blood Nerve Barrier (BNB)

The morpho-functional organization of the blood/tissue barriers, BBB and BNB in particular, has strong affinity elements based on the specialization of the endothelial cells that compose them. However, in situ observational studies, conducted on the peripheral nerves of humans and rodents, have shown significant differences in the various districts that obey the different needs of the tissues that the barriers are called to modulate. In particular, there is evidence that it is possible to find a lower number of fenestrations in BNB endothelial cells, compared to those observed in the BBB endothelium. This implies that the BNB has more restrictive barrier properties than the BBB [8]. This is also confirmed by the greater presence of intercellular junctions both adherent (AJ) and tight (TJ). Furthermore, it has been documented that the proteins associated with the junctions are also more numerous. In particular, BNB endothelial cells express claudins (claudin 1, claudin 2, claudin 5, and claudin 19), occludins (ZO-1, ZO-2), and junctional adhesion molecules (JAM-A) [9].

This has significant functional implications. In fact, the passage of leukocytes through the microvascular endothelium is necessary for the immunosurveillance of the tissues, as well as for the response to lesions, inflammation, or infections (Table 1). However, excessive passage of leukocytes has been considered a pathological sign of autoimmune neuropathies. In particular, an increase in the passage of leukocytes was associated with inflammation of nerves, thereby causing alterations in the structure of the BNB. Under these conditions, reduction in claudin 5 expression and specific alterations in ZO-1 localization have been observed and this represents an early biomolecular mechanism involved in BNB alterations in peripheral neuroinflammatory diseases [10].

Due to the greater restrictive property of the BNB, specific transporters must be present on the luminal and abluminal surfaces of the endothelial cells in order to facilitate the directional influx of solutes, nutrients, and macromolecules and the outflow of metabolic or xenobiotic waste. The enzyme alkaline phosphatase (AP) can be considered an ionic transporter of the capillary endothelium in the peripheral nerves of mammals; in particular, it plays a crucial role in these processes, since it participates in the transfer of phosphate groups and acts as an ionic membrane pump capable of preserving ionic concentrations within the BNB endothelium [14].

At the BNB level, many transporters facilitate the passage of glucose, amino acids, and lipids from the bloodstream to endothelial cells (Table 2).

The main glucose transporter in the body—highly expressed by human BNB endothelial cells—is Glucose transporter-1 (GLUT-1), which transports D-glucose into the endothelium to be used as the main source of energy, under physiological conditions. On the other hand, the monocarboxylate transporter 1 (MCT-1) is similarly expressed in human endothelial cells of the BNB and regulates the influx of lactate, which represents an energy source under anaerobic conditions as well us under starvation states, thereby allowing a physiological outflow of its metabolic bio-products [15]. Furthermore, the creatine transporter (CRT) is also expressed in BNB vascular endothelium. Creatine, in particular, supplies high-energy phosphate groups necessary for the production of ATP in order to guarantee the normal energetic function of the peripheral nerve [18]. Finally, the ABC transporters (ATP bond box) guarantee the outflow of xenobiotics in the BNB endothelium; these transporters include P-glycoprotein (MDR1), which eliminates chemotherapeutic, antineoplastic, antimetabolite, and heavy metal substances. Moreover, it is fundamental for the protection of peripheral nerves from external factors and potentially toxic tissue metabolism intermediates. Finally, it can represent potential biomarkers for detecting early changes of BNB functionality in peripheral nerve dysfunction [19].

Thus, based on previous mechanisms, several endothelial dysfunctions can actively cause vascular damage and the progression of various nerve ending diseases. Among them, the endothelial alterations associated to cardiovascular disease imply alterations of nerve/blood interface and disruption of BNB [21,22,23].

## 3. Alterations in the Production of Nitric Oxide at the BNB Level

The correct functioning of endothelial cells in vascular tissues as well as at the level of BNB is mainly expressed by their capacity to regulate the basal release of soluble mediators, such as nitric oxide (NO). In particular, the pulsed generation and release of NO in response to physiological stimuli (i.e., shear stress, bradykinin, etc.) represents the major component for assessing endothelial cell functionality. On the other hand, it is well known that an impairment of NO release leads to the so-called endothelial dysfunction [24].

Chemically, NO is a very reactive and unstable soluble gas with a short half-life (1–10 s); based on this characteristic, NO is preferably located near the cells that produce it, acting as an autocrine/paracrine factor. Since NO is characterized by small dimensions and the absence of net charges, it spreads freely through biological membranes without any specific transport system. In its radical form, NO is oxidized, under physiological conditions, to NO_2_ and NO_3_ [25]. NO is constantly synthesized in endothelial cells as a result of the conversion of l-arginine into the amino acid l-citrulline, a reaction which is catalysed by the enzyme nitric oxide synthase (NOS) [26]. There are three isoforms of the enzyme NOS: (1) Endothelial isoform (eNOS) is present in endothelial cells in osteoclasts, in osteoblasts and in renal mesangial cells. Endothelial NOS strictly depends on calcium ions, since it is dependent on the calcium-calmodulin system [27]. The basal release of NO is pulsed and occurs at nanomolar concentrations; (2) neuronal isoform (nNOS) is found constitutively in the nervous system, in which the formation of NO modulates the release of many synaptic neurotransmitters [28]; (3) inducible isoform (iNOS), observed in macrophages, neutrophils, and other inflammatory cells [29], induces the release of micromolar concentrations of NO.

Despite this strict classification, NO can also be produced in other types of cells, including skeletal muscle cells, cardiac cells, and platelets [30].

In the human genome, there are three different genes for the three NOS isoforms on chromosome 7 (eNOS), 12 (nNOS), and 17 (iNOS), respectively; they share a high degree of affinity that suggests the existence of a common ancestral gene [31].

The NOS commonly have a dimeric form in their active state: in particular, they consist of a domain of oxygenase and reductase; the reductase domain is linked to the oxidase domain by a calmodulin-binding sequence. The N-terminal oxygenase domain consists of the binding site for the heme group, tetrahydrobiopterin (BH4), and l-arginine. In particular, BH4 is an important cofactor of NOS, as it is involved in the transport and transfer of electrons in the reaction that it catalyses. In contrast, the C-terminal reductase domain corresponds to a specific site for calmodulin, flavin adenine dinucleotide (FAD), flavin adenine mononucleotide (FMN), and nicotinamide adenine dinucleotide phosphate (NADPH). The oxygenase and reductase domains are physically and functionally connected; however, only the binding with BH4 and calmodulin contributes to the stabilization of the enzyme. Flavin cofactors accept electrons from NADPH and transfer them to the heme, which is considered the final step in electron transfer. Furthermore, the binding to calmodulin increases the rate of electron transfer from NADPH to flavones of the reductase domain and from the reductase domain to heme [31]. NOS isoforms can be controlled at the transcriptional level through alternative mRNA splicing and variations in covalent bonds [32].

At post-translational level, the regulation of the eNOS activity depends on the specific site of phosphorylation. Specifically, the phosphorylation of eNOS-Ser^1177^ is activated in response to mechanical stimuli, such as endothelial shear stress or humoral factors. This, in turn, leads to the binding to Ca^2+^/calmodulin, thus increasing of eNOS activation. Then, the activity of eNOS is enhanced after Ser^633^ phosphorylation. This residue is located in the CaM autoinhibitory sequence of eNOS and, consequently, its phosphorylation determines eNOS activation and increased NO levels, without requiring an intracellular Ca^2+^ increase.

Shear stress and high density lipoproteins (HDL) increase the phosphorylation on eNOS-Ser^114^ while the eNOS agonists such as VEGF, statins, or bradykinin increase the phosphorylation of eNOS-Ser^615^. Several studies showed that the phosphorylation at this residue leads to an increased sensitivity of the enzyme to Ca^2+^/CaM, although the effect of the phosphorylation of this site results controversial.

Further evidence shows that, the phosphorylation of eNOS at Thr^495^ residues leads to a reduction of activity of this enzyme; in particular, in vitro studies conducted on cultured endothelial cells demonstrated the involvement of the protein kinase C in the phosphorylation at eNOS^-^Thr^495^ [33] (Figure 1).

Another level of NOS regulation is related to Arginase-1 activation state. Indeed, in condition of Arginase 1 hyperactivation, there is a reduction in the amount of l-arginine, which represents the substrate of NO synthesis. Consequently, the reduction of NO levels causes an increase in ROS [34,35] (Figure 2).

Moreover, in the production and release of NO from endothelium many agonists are involved; among them, acetylcholine, histamine, thrombin, serotonin, bradykinin, substance P, isoproterenol, and norepinephrine bind to specific receptors on the membrane of endothelial cells, resulting in a rapid release of NO [36]. More specifically, the binding of the agonist receptor leads to an increase in the concentration of cytosolic calcium released from the endoplasmic reticulum, with the subsequent binding and activation of eNOS. In this scenario, the NO production is strictly dependent on the amount of available calcium; in fact, at low concentrations the calcium-calmodulin complex dissociates from the enzyme, causing its deactivation [37].

The new synthesized NO leaves the endothelial cells and diffuses into adjacent smooth muscle cells, where it stimulates the soluble guanylate cyclase (sGC) enzyme, involved in the increase of cyclic guanosine monophosphate (cGMP). The accumulation of cGMP, in turn, reduces muscle tension on one side and, on the other, minimizes the release of calcium ions from the endoplasmic reticulum. The combination of these two effects decreases the contraction of smooth muscle cells and, therefore, leads to vasodilation; finally, cGMP effect can be restored or neutralized by the action of the enzyme phosphodiesterase type 5 [38]. Thus, the constitutive release of NO contributes to control and regulate the vascular tone. However, NO has several other functions, such as anti-inflammatory properties, that manifest through the inhibition of the synthesis of cytokines and the inhibition of specific molecules responsible for the recruitment of inflammatory cells, thus allowing their passage through the vessel walls [39]. In addition, NO inhibits platelet aggregation and cell migration [40]. The formation of NO can be also inhibited when occurs in conjunction with certain disease states, such as hypertension, diabetes mellitus, and dyslipidemia or in particular predisposing conditions, such as aging and smoking [41].

## 4. Endothelial Dysfunction in BNB-Related Disorders

Endothelial dysfunction at the interface between blood and peripheral nerve is involved in the onset of many disease states via different mechanisms which include: (1) The reduction of NO; (2) an increased expression of pro-inflammatory factors; and (3) the modification of endothelial permeability [42].

The first mechanism is associated with a low bioavailability of NO. The role of NO in modulating the vascular tone was evidenced by experiments focused on the inhibition of its synthesis. In fact, the administration of N^G^-monomethyl-l-arginine (l-NMMA), L^G^-nitro-l-arginine (l-NA) and N^g^-nitro-l-arginine methyl ester (l-NAME) inhibits the release of NO from endothelial cells. Furthermore, the administration of these inhibitors leads to an increase in blood pressure of about 30 mm Hg in rats and mice [43,44]. Similar results have also been observed in human studies [45]. The vessel adaptive response to NO release mediated by EC also depends on NO concentration. Indeed, low levels of NO induce a lower degree of vasodilation; in contrast, drastically reduced levels of NO determine vasoconstriction.

The NO reduction also derives from the accumulation of Reactive Oxygen Species (ROS) involved in the onset of oxidative stress through different mechanisms.

The induction of ER stress in endothelial cells involves an increased expression of endothelin-1 and a reduction of eNOS. On the contrary, it has been demonstrated that counteracting ER stress, a better activity of eNOS and a greater vascular relaxation were observed. There are several mechanisms underlying the correlation between ER stress activation and endothelial function. One of these is the insulin resistance that causes an altered production and signalling of NO. Other mechanisms include an increase in oxidative stress, due to the reduced bioavailability of NO, endothelial cell apoptosis, and inflammation. In conditions of non-reversible ER stress, such as under increased ROS levels and impaired calcium homeostasis, the endothelial cell activates proapoptotic signals (i.e., JNK/p38 or caspase-12). In this context, the inositol requiring protein-1 IRE-1 stimulates JNK and P38 (MAPK) through the apoptotic signal ASK1. In turn, ASK1 inhibits eNOS by reducing phosphorylation at Ser^1177^ site and causes NO deficiency. On the other hand, enhanced ROS and dysregulated calcium levels directly promote apoptosis through the activation of caspase-12 (Figure 3) [46,47].

Under physiological conditions, there is a balance between the endogenous production of free radicals and their neutralization by antioxidant systems. However, in the context of increased ROS production, a dramatic reduction of the ability of endogenous antioxidant moieties to remove them has been found to occur. Once the ROS are generated, they react with organic and inorganic molecules, thus producing other radicals with a series of chain reactions. The change in the redox state results in a reduced eNOS activity, with a consequent inhibition of NO formation, and a greater consumption of NO by ROS [48] (Figure 2).

The superoxide anion (O_2_^−^) appears the most dangerous species among ROS, since its accumulation implies a rapid reaction with NO, leading to peroxynitrite formation. In this case, NO reacts with O_2_^−^ and leads to the formation of peroxynitrite (ONOO^−^), a strong oxidizing molecule able to alter the structure of biological macromolecules, including those involved in the NO pathway [49].

Furthermore, ROS modify the permeability of the endothelium, thus allowing the overcome of toxins through the barriers localized between blood and tissues in order to reach the target tissues [50]. This evidence is also supported by different experimental observations; in particular, in patients with high cardiovascular risk treated with antioxidant substances, the bioavailability of NO, and the consequent endothelium-dependent vasodilation are improved [51]; moreover, in those patients with a reduced contribution of antioxidants, a lower bioavailability of NO together with an increased vasoconstriction were observed [52]. Furthermore, it has also been shown that the excessive amount of ROS can alter mitochondrial oxidative phosphorylation, with consequent discharges of calcium deposits [53]. These changes, representing the main steps in endothelial dysfunction, are also accompanied by the enhancement of several pro-inflammatory factors. This process is triggered by mechanical or biochemical stimuli that the endothelium cannot address and overcome with adequate responses to stressful stimuli. Indeed, under chronic stressful conditions, the endothelial cells alter their phenotypic aspect by up-regulating the expression of cell adhesion molecules (CAM), such as intercellular adhesion molecule 1 (ICAM-1), the adhesion molecule of vascular cells (VCAM-1) and selectin E [12]. These molecules are expressed on the plasma membrane of endothelial cells and increase the affinity of the leukocytes and the weakening of the barrier, thus causing the leukocyte diapedesis in the peripheral tissues. Furthermore, an increase in pro-inflammatory C-reactive protein has been shown to occur [54]. Recently, special attention has been paid to an endogenous inhibitor of the eNOS isoform, the asymmetric dimethyl-arginine protein (ADMA), which has been shown to be a potential biomarker for endothelial dysfunction. Indeed, it has been shown that the plasma levels of ADMA are directly correlated with NO amount [55]. Many other molecules, such as inflammatory cytokines, have been mentioned as indicators of endothelial damage [56] (Figure 4).

Besides NO, endothelial cells are also the source of numerous other factors in physiological conditions or as a consequence of a vasoactive stimulus. These mediators regulate the vascular tone and the responsiveness of the endothelium by exerting different effects. The Endothelin-1 possesses inflammatory and proliferative action, inhibits eNOS, reduces NO release, and antagonizes NO action. Prostacyclin and the endothelium-derived hyperpolarizing factors (EDHF) induce vasodilatation and have an antihyperproliferative effect. Several studies have also observed that EDHF may act as vasodilator in the presence of an altered bioavailability of endothelial NO [57,58,59].

An increase in pro-inflammatory molecules is also associated with increased endothelial permeability, leukocyte adhesion, and monocyte migration [13]. Despite the different nature of these two mechanisms underlying endothelial dysfunction, they can occur simultaneously: the increase in intracellular ROS concentration, in fact, activates the nuclear factor NF-kB, thus causing an increase in the transcription of proinflammatory genes and expression of different adhesion molecules [56].

Moreover, also the occurrence of peripheral neuroinflammation leads to early changes in the BNB, being associated to its partial or total destruction; these events are both accompanied by changes in endothelial permeability [13]. Indeed, accumulating evidence shows that many pathophysiological events leading to injury of peripheral nerve—as a result of trauma, aging or osteoarthritis—favours the infiltration of different categories of white blood cells. In particular, in the affected area, lymphocytes can capture and degrade the antigen, directly through the specific receptors or indirectly through the antigen-presenting cells as mediator. The presence of immune cells such as lymphocytes and macrophages, which overall are named “immunocytes” or “immunocompetent cells”, is the basis for the characterization of a neuroimmune process which represents the origin of the development of several peripheral nerve diseases. In particular, in vivo experiments have shown that the presence of inflammatory cells in the area of nerve lesion occurs early (3 hours) after the onset of damage to the peripheral nerve and that neuroinflammation, expressed by infiltration of injured nerves by immunocompetent cells, persists for up to two months after damage [60]. In addition to the activation and infiltration of immune cells, an increased synthesis of pro-inflammatory mediators such as interleukin 1 (IL-1β), interleukin 6 (IL-6), and tumour necrosis factor (TNF-α), across to injured nerve fibers, has also been shown to occur [61]. To further confirm the existence of a specific inflammatory process accompanying peripheral neuropathy, an increase in Toll-like receptors (TLRs) expression was also found (Table 3).

TLRs are transmembrane receptors able to recognize the molecular characterization of pathogens or microbes. Structurally, they are all integral glycoproteins with an extracellular structure containing leucine residues and characteristic cysteine traits involved in the bond. In the cytoplasmic area, there are some tails that contain a cellular activation domain. When a pathogen passes through the host barriers, it is recognized by the TLRs that immediately activate the sentinel cells of the immune responses. In mammals, TLRs are expressed on the membranes of leukocytes, macrophages, Natural Killer cells, endothelial cells, and epithelial cells. TLRs can also dimerize or heterodimerize, thus improving specificity for other molecular profiles [63]. Some data can be found in the literature suggesting that damaged neurons, after injury occurring in peripheral nerve, increase the expression of some TLRs, thus triggering an immune response and the onset of alteration or destruction of the BNB [64].

The integrity of BNB endothelial cells is an event which contributes in counteracting the development of neuroimmune disorders accompanying peripheral nerve dysfunction. In fact, it has been clearly assessed that in order to guarantee the action of the BNB as a restrictive barrier, the endothelium can contain the expression of the proteins of the tight junctions, of the adherent junctions, of the junctional adhesion molecules (JAM-A), of cadherins and of β-catenin. Furthermore, the expression of claudins-1, 5, 12, 19 as well as occludins (ZO-1; ZO-2) [11] was detected in primary and immortalized endothelial cells. In contrast, biochemical and functional changes found in BNB endothelium represent an event which occurs at the early stages of peripheral nerve injury characterized by neuroinflammatory processes (Figure 5). In particular, in the event of peripheral nerve inflammation, there is a reduction in tight junction proteins at the endothelial level. Furthermore, it has highlighted a decrease in the expression of occludin and claudin 5, which are essential for the endothelial component and that act as markers of several changes in endovascular permeability. This is confirmed by experiments carried out in male rats, after induced damage following sciatic nerve constriction, which showed lower levels of mRNA for occludins and for claudin 5 [67]. Overall, these events lead to the desensitization of the peripheral nerve, to the alteration of the BNB, and to the onset of neuropathic pain [68].

Some examples of alterations affecting the peripheral nerves and involving the endothelium will be explained below. In particular, we will focus on diabetic neuropathy, erectile dysfunction, and inflammation of the sciatic nerve.

## 5. BNB Dysfunction in Diabetic Neuropathy

Diabetes mellitus is a disease now considered to be an epidemic disorder due to its increasing prevalence among the world population, as evidenced by the International Diabetes Federation. This disease is mainly due to various factors mainly depending on the intake of dietary excess calories as well as on a sedentary lifestyle [69]. In this context, endothelial dysfunction is a disorder that is constantly associated with both type I (insulin-dependent) and type II diabetes, characterized by high concentrations of serum insulin and by insulin resistance, occurring at the early stages of the disease. This alteration can be found at level of both the CNS and PNS [70].

It has been recognized that about half of diabetic patients suffering from peripheral neuropathies undergo different stages of diabetic neuropathies [71]. The most frequent complications of diabetes mellitus are represented by symptoms ranging from subclinical forms of neuropathy to clinical and, often, irreversible forms of peripheral sensorimotor neuropathies [72]. These forms are characterized by the loss of symmetrical sensitivity in the feet, legs muscles, and calves [73], often accompanied by pain, tingling, paraesthesia, and numbness.

The progression of the disorder aggravates these symptoms, leading to motor nerve dysfunction and total insensitivity in the toes, ankles, and calf muscles. The progressive loss of sensitivity in the lower limbs culminates over time with the loss of balance and difficulty in maintaining the upright position [74]. Recent studies have also highlighted the onset of peripheral neuropathies in the upper limbs and a close correlation between weakening of the hands and diabetes mellitus. In addition, a direct relationship between sensitivity and glycaemic load as well as a clear connection between the onset of damage and aging were revealed [73].

Endothelial dysfunction occurs in specific clinical stages of diabetic neuropathy [75] independently on the serum glucose levels; moreover, it derives from the inhibition of some glycolytic enzymes, including D-glyceraldehyde 3-phosphate dehydrogenase, with consequent accumulation of many intermediates, which trigger different inflammatory pathways, characterized by the production of metabolites such as polyols, hexosamines, and protein kinase C [75]. Each of these pathways is capable of damaging the peripheral nerve; furthermore, the resulting imbalance between mitochondrial redox state and ROS formation results in the dysfunction of endothelial cells in that specific district.

In particular, in the polyol pathway the excessive glucose is transformed into sorbitol by the aldose reductase enzyme, thus causing an osmotic damage with the release and loss of myo-inositol and taurine. The reduction of myo-inositol alters the physiology of the nerve, since it is an essential component and guarantees the correct functioning of the Na^+^/K^+^ pump depending on the ATP; on the other hand, dysfunction of endothelial cells occurs due to the reduction of NADPH, which is necessary for NO production and ROS accumulation [76].

Furthermore, the hexosamine pathway is involved in the increased production of fructose-6-phosphate, which favours the inflammatory process in endothelial cells and in the peripheral nerve endings [77].

Finally, a further potential mechanism contributing in endothelial dysfunction correlated to diabetes mellitus is represented by the activation of the protein kinase C pathway which can lead to the consequent increased glycolysis and to the accumulation of diacylglycerol (DAG). This activates neuronal protein kinase C, causing nerve damage and increasing the expression of pro-inflammatory factors in endothelial cells [78]. Therefore, all three pathways involved in the onset of diabetic neuropathy involve endothelial dysfunction resulting from an increase in ROS as well as from an increased inflammatory process.

A further contribution to the complex machinery which associates diabetes mellitus to BNB alterations is represented by endoplasmic reticulum dysfunction.

The endoplasmic reticulum is an organelle of the eukaryotic cell constituted by a series of membranes that depart from the nuclear envelope. It has many functions, including the synthesis, maturation, and folding of secretion proteins, alongside with the synthesis of lipids and calcium ion reserves. In the case of accumulation of misfolded proteins following reticular stress, a defensive pathway involving the organelle, known as “unexplained protein response” (UPR), is activated. At the molecular level, there are three transmembrane proteins involved in UPR activation: (1) Protein-kinase-like ER Kinase (PERK), (2) IRE-1, and (3) activating transcription factor-6 (ATF6). The UPR relieves the reticulum stress in different ways: it triggers the transcription of the genes related to the proteins chaperones intended to improve the “folding” of proteins; it slows down the synthesis of new proteins in order to avoid the formation of additional misfolded proteins; finally, it triggers the “endoplasmic-reticulum-associated protein degradation” (ERAD) with the aim of eliminating the misfolded proteins from the lumen of the organelle [79].

In the endothelial dysfunction associated to the occurrence of diabetes mellitus, the endoplasmic reticulum is actively involved. In fact, an abnormal accumulation of misfolded proteins is observed, together with the activation of PERK, IRE-1, and ATF6 [80]. As a result of high glucose serum concentrations, in fact, the UPR activation increases the ROS production, probably induced by the NADPH oxidase enzyme [81]. Moreover, the accumulation of ROS also reduces the calcium ion levels within the reticulum, while increases its cytoplasmic concentration. In turn, the latter process stimulates the alteration of the electronic transport at mitochondrial level, being followed by the release of cytochrome c and, at the late stage, to the apoptosis of the endothelial cells [82,83].

## 6. BNB Involvement in Erectile Dysfunction

Some epidemiological studies suggest that erectile dysfunction (ED) is quite common in men and affects 50% of individuals aged 40 to 70 years old [84]. The main cause of ED is generally a vascular dysfunction of the penile arteries (80% of cases) [85]. However, the finding of a major atherosclerotic process in the penile vessels is present only in the advanced stages of ED. Conversely, several studies, aimed to analyse the endothelial function in disorders associated with erectile dysfunction, have shown an altered endothelium in the early stages of the pathology. These changes are mainly associated with an alteration of the so-called “vasa nervorum” which, ultimately, subtend to an early dysfunction of the microcirculation at the BNB level.

Additional risk factors for the onset of ED are aging, obesity, hypertension, diabetes, hyperlipidemia, metabolic syndrome, and vascular diseases. In fact, several studies have shown that the onset of ED increases by 49% in patients with type 1 diabetes and 34% in patients with type 2 diabetes [86,87]. Furthermore, ED can be predictive for the onset of cardiovascular diseases, since more than half of ED men undergo cardiac symptoms with alterations in the coronary arteries [88]. Other modifiable risk factors are related to ED, including smoking, a sedentary lifestyle, the wrong diet, overweight, and alcohol abuse [89].

Both tobacco and passive smoking greatly increase the likelihood of developing ED; indeed, a recent study showed that the overall ratio for ED development was 1.34 among non-smokers and 1.61 in smokers [90]. Furthermore, additional evidence shows that erectile function has significantly improved in ex-smokers after about a year [91].

Positive effects on erectile function can derive from regular exercise, which reduces the probability of developing ED, with an overall ratio from 0.63 to 0.43 [92]. Epidemiological studies also suggested that the overweight or obesity conditions are associated with a greater risk of developing an ED of 40% [93]. In particular, the preferential assumption of foods, legumes, fruit, vegetables than red meat, dairy products, foods and beverages rich in sugar have a lower risk of developing ED than those with an unbalanced diet [94]. In this regard, recent studies have shown how supplementation with natural extracts rich in polyphenols, and therefore with high antioxidant power, such as the bergamot polyphenolic fraction (BPF) given in adequate concentrations, reduce endothelial dysfunction and ED in patients suffering from type 2 diabetes [86]. Finally, it should be remembered that the consumption of alcohol in large quantities and for long periods is able to determine an early alteration of the endothelial function associated with BNB. This implies an increase in the possibility of developing ED equal to 25–30% compared to subjects not drinkers [95,96].

In general, therefore, the element shared by the pathological and modifiable risk factors of ED is the endothelial dysfunction, correlated with an early reduction of BNB functions [97]. In fact, it is known that during the erection of the penis, the muscles of the cavernous bodies of the penis relax, while the arterial flow increases and the venous flow decreases. This mechanism is regulated by some nerves which, through transmitters and modulators, transform the phenomenon of erection into a neurovascular process. A key role in the erection process is played by NO, which controls the relaxation of the corpora cavernosa. NO is produced by endothelial cells of the penile arteries by means of the eNOS and nNOS isoforms, thereby promoting relaxation of the cavernous body through the formation of cGMP [98]. In the case of endothelial dysfunction, found in ED, the availability of NO and, consequently, of guanylate cyclase [99] is reduced, and the effect of this modulation is followed by vasoconstriction and the increased production of vascular inflammatory cytokines.

This happens as a result of the altered dimerization of the NOS enzyme and the subsequent loss of function, an effect accompanied by overproduction of free radical species [100].

Furthermore, in the development of endothelial dysfunction that underlies the onset of ED symptoms, a greater expression of the catalytic subunit of the NADPH oxidase that is the main producer of reactive oxygen species in the endothelium and in vascular smooth muscle cells, occurs [101,102]. The catalytic subunit of NADPH oxidase (NOX) catalyses the formation of superoxide anion through the transfer of an electron, transferred from NADPH or NADH, to oxygen [103]. The accumulation of free radicals causes enormous damage to endothelial cells, thus prompting the immune system to respond to warning signals through an increased production of pro-inflammatory cytokines, such as tumour necrosis factor alpha (TNF-α), interleukin-6 (IL-6), and interleukin-8 (IL-8). Thus, the accumulation of ROS and the production of pro-inflammatory cytokines cause structural damage of the endothelium leading to the development of vascular and tissue damage [103]. This effect represents the major element of vulnerability of the BNB that loses its filter function and, at the level of the penile nerves, determines an alteration of the nerve endings with reduced release of neurotransmitters that block the process of erection.

## 7. BNB and Neuropathic Pain

Peripheral nerves injuries (PNI) accompanied by neuropathic pain are common alterations: In fact, about 1% of the population aged 70 or more is affected by PNI. PNI can lead to the demyelination or partial axonal degeneration, which culminate, in both cases, with acute pain, muscle paralysis, loss of sensitivity, and permanent disability [104]. Neuropathic pain is defined as “pain caused by a lesion or disease of somatosensory system”-it is not a single pathology but a set of different diseases that is associated to numerous symptoms.

It is possible to distinguish two main types of symptoms in neuropathic pain, which differ for pathophysiological and neurobiological properties: Allodynia and hyperalgesia. In allodynia, pain is generated by a stimulus that normally does not cause pain, while in hyperalgesia pain is provoked by a stimulus that normally causes pain. Despite pain is subjective in each individual, it includes different factors: Sensory, emotional, and cognitive [105]. The most common experimental model used in laboratory research to reproduce PNI and to develop neuropathic pain, relies on rodents whose lesions are generated by an induced alteration of the sciatic nerve [106]. The sciatic nerve is one of the main nerves in the lumbar area and its inflammation is very painful, as it stems from the combination of several nerves in the lumbar spine. The main cause of the inflammation of the sciatic nerve is the disc herniation (L5-S1, or, more rarely, L4-L5.) Other causes include ageing and the consequent reduction of the intervertebral spaces, arthritis, hyperlordosis, and traumas. Acute pain affects the loins, the gluteus, the back of the thighs and the legs, and the toes. The most serious cases are accompanied by the limb paresis. Experimentally, the injury to the sciatic nerve can result from the mechanical compression, and this experimental model helps to detect the possible changes in the nerve at proximal and distal level with respect to the injury. The compression of the sciatic nerve may be due to the transection or the crushing of the epineurium, which is the outermost part of the peripheral nerve. The transection is the method mainly used to study neuropathic pain and leads to a transient injury to the sciatic nerve, which is evidenced only by functional changes [107]. On the other hand, the injury to the sciatic nerve lesion, deriving from the crushing, causes a permanent anatomical injury, as demonstrated by studies on the repair and regeneration of the nerve at cellular and molecular levels [108]. The nerve regeneration, as a result of an injury to the sciatic nerve, can be a predictive data particularly useful at pre-clinical level [109]. A more severe clinical condition is caused by chronic denervation, which is the result of severe traumas. In this case, a permanent disconnection between neurons and their targets is observed, with the consequent loss of sensory and motor functions. Experimentally, chronic denervation can derive from the complete cutting off of the sciatic nerve, without any partial surgical reconstruction [110]. This model is not used for the study of the preservation of the sensory function, rather than the search of strategies able to prevent denervation [111].

In general, the injury to the peripheral nerves causes morphological and phenotypic changes in the nerve fibres, thus leading to hyperalgesia and allodynia [112]. More specifically, the alterations in the sciatic nerve may result in the breaking of the BNB, neuronal inflammation, fibrosis, and diffuse local demyelination [113]. These alterations of sciatic nerve are also associated with increased expression of many pro-inflammatory cytokines, including IL-1α, IL-1β, TGF-β e IL-8 [114], an effect associated with the onset, the development and the persistence of neuropathic pain [115].

Alongside with the induction of an inflammatory process, the persistence of neuropathic pain within changes occurring in the sciatic nerve is also due to the production and the accumulation of ROS [116]. In particular, a significant increase in the glutathione levels (GSH) as well as in the activity of glutathione peroxidase (GPx) and glutathione transferase (GST) enzymes, respectively involved in the reduction of hydrogen peroxide and in the detoxification of xenobiotics and lipid peroxidation by-products [117], has been observed. The increase in the GSH level has been intended as a substrate for the activity of the GPx enzyme, while the increase in the activity of the GPx enzyme corresponds to the increase in the hydrogen peroxide, as observed in many experimental settings. Moreover, the treatment with anti-oxidants has been shown to reverse or prevent several changes caused by the injury to the sciatic nerve [118].

Moreover, during the development of sciatic nerve, a higher expression of the nNOS enzyme is reported. The consequent increase in the NO production can easily lead to the formation of peroxynitrite. On the basis of this evidence, it has been observed that the anti-oxidant treatment is able to reduce hyperalgesia in experimental settings of neuropathic pain induced by injuring the sciatic nerve [119].

The involvement of endothelial dysfunction and subsequent BNB damage in experimental models of sciatic nerve injury is confirmed by evidence showing that the nerve ligature is associated with changes in the expression of intercellular junction proteins [111,120,121]. In particular, in this experimental model, the permeability of the BNB was assessed, at endothelial level, after the exposure to Evans Blue. After 3 hours from the damage and especially after 6 hours, the results showed the migration of Evans Blue outside the blood flow, thus confirming an increased endothelial permeability. These results have confirmed the occurrence of an altered regulation of occludins, claudins, gap junctions, and caderins [122], accompanied by a prominent infiltration of immune cells as well as by an enhanced production and accumulation of inflammatory cytokines [67]. Thus, a clear association exists between sciatic nerve injury and BNB alterations, which represents the basis for the development of neuropathic pain [123].

## 8. Conclusions

BNB pathophysiology is strictly connected to endothelial dysfunction, which can also represent a direct cause of peripheral nerve damage. In the present review, we have investigated on the possible cause of the BNB permeability, identifying oxidative stress and inflammation as the leading mechanisms of endothelial dysfunction. As several peripheral neuropathies are characterized by the impairment of the endothelial cells, we hypothesize that a strategy aimed to restore early BNB permeability remains a valid therapeutic approach to counteract oxidative- and inflammation-driven nerve injury. On the basis of these observations, the identification of further molecular mechanisms involving in BNB disruption can open novel and important perspectives also in the management of neuropathic pain.

## Figures and Tables

**Figure 1 ijms-20-03022-f001:**
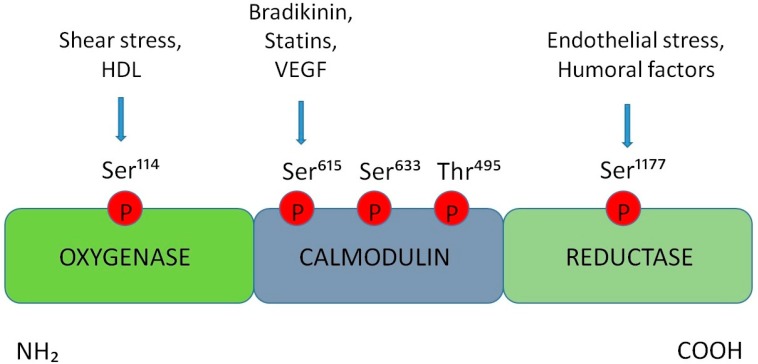
Nitric oxide synthase (NOS) regulation. Schematic representation of the phosphorylation sites on the endothelial isoform (eNOS) enzyme and of the mechanical and humoral factors involved. Shear stress and HDL increase the phosphorylation on eNOS-Ser^114^ site; the eNOS agonistic VEGF, statins, and bradykinin increase the phosphorylation of eNOS-Ser^615^. Moreover, the phosphorylation of eNOS-Ser^1177^ and eNOS-Ser^633^ leads to an increase of eNOS activity, on the other hand the phosphorylation of eNOS-Thr^495^ leads to a reduction of the activity of this enzyme.

**Figure 2 ijms-20-03022-f002:**
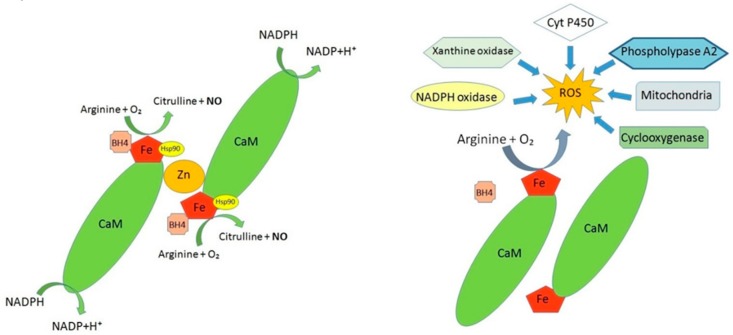
NOS activity. Under normal condition calcium is associates with calmodulin to activate the enzyme eNOS which produces NO from its precursor l-arginine forming l-citrulline. In this context the enzymatic reaction generating NO involves the transfer of electrons from NADPH, via the flavins to the heme and consequently the substrate l-arginine is oxidized to l-citrulline and NO. To efficiently produce NO, eNOS must effectively coordinate the binding of multiple substrates and cofactors such as Tetrahydrobiopetrin (BH4). Disruption of this highly coordinated catalysis (uncoupled eNOS) can result in the production of superoxide and peroxynitrite. Other sources of ROOS are mitochondria, phospholipase A2, Cyclooxygenase, Cyt P450, xanthine oxidase, NADPH oxidase.

**Figure 3 ijms-20-03022-f003:**
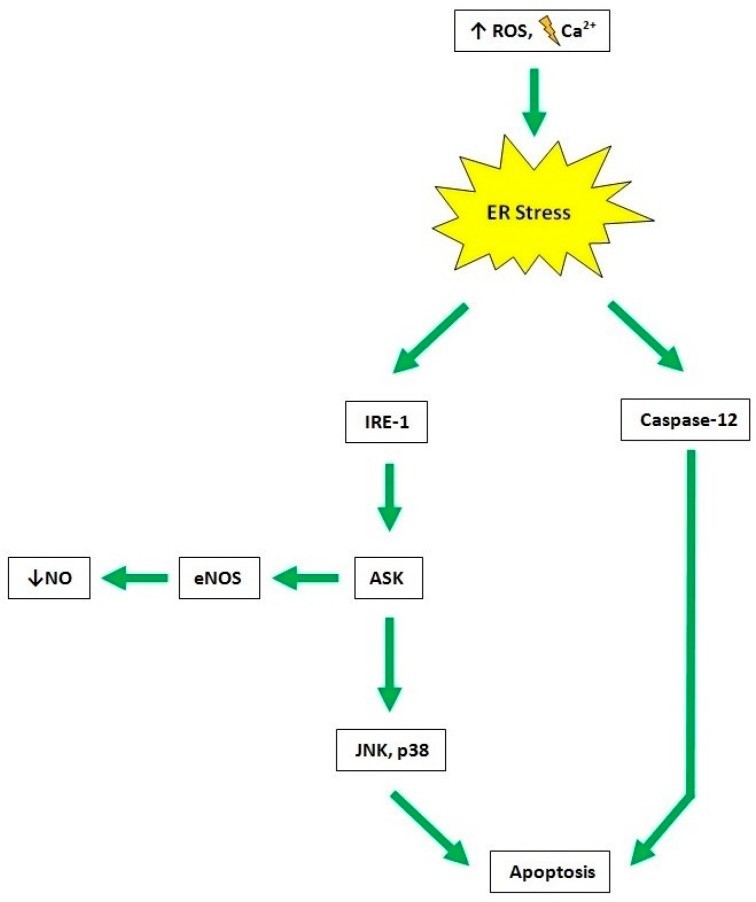
ER stress. The increased ROS levels or insulin resistance lead to ER stress in endothelial cells. While insulin resistance altered the production of NO, the reduced bioavailability of NO induces an increase of oxidative stress leading to an impaired calcium homeostasis and to the activation of proapoptotic signals such as JNK/p38 or caspase-12 promoting apoptosis. IRE-1 through the apoptotic signal ASK1 stimulates proapoptotic signals. Moreover, ASK1 decrease eNOS levels and causes NO deficiency.

**Figure 4 ijms-20-03022-f004:**
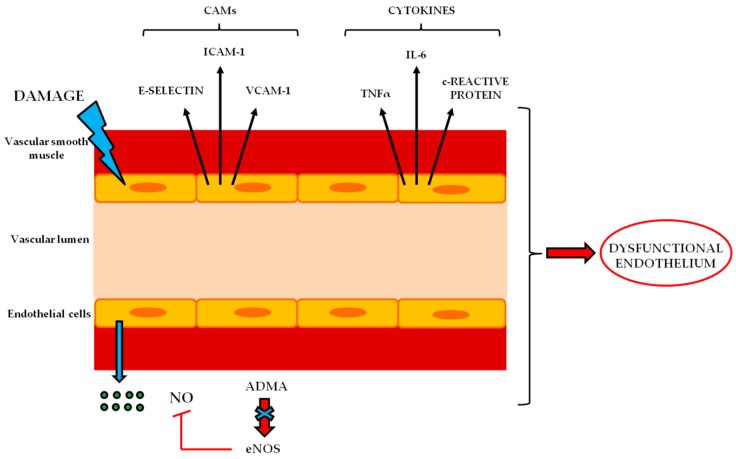
Dysfunction of endothelium. The injury of endothelial cells is accompanied by neuroinflammation due to increased expression of pro-inflammatory factors.

**Figure 5 ijms-20-03022-f005:**
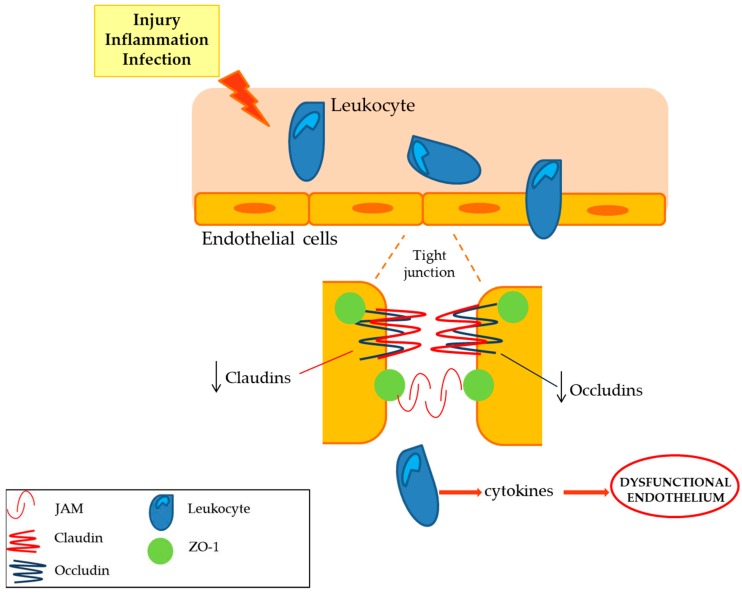
Endothelium dysfunction and leukocytes infiltration. During pathological conditions, such as inflammation, infection, and injury of peripheral nerve, an alteration of endothelial intercellular junction proteins occurs. A reduction of tight junction proteins, such as claudins and occludins, determines a leukocytes infiltration.

**Table 1 ijms-20-03022-t001:** Components of Blood Nerve Barrier (BNB) intracellular junctions.

BNB Components	Role in BNB	Expression in BNB-Related Disorders
Claudins (claudin 1, claudin 2, claudin 5 and claudin 19)	Tight junction that limits the cellular permeability. They modulates the passage of leukocyte, regulating the immunosurveillance of the tissues [10].	The expression of claudin 5 is decreased while the claudin 1, 2 expression is not affected in peripheral nerve inflammation. Deficient claudin 19 mice showed a deficit in PNS [10,11].
Occludins (ZO-1, ZO-2)		The expression of occludins is decreased in peripheral nerve inflammation, whilst ZO-1 and ZO-2 localization is altered [10,11].
Cell adhesion molecules		The expression of intercellular adhesion molecules, such as ICAM -1, VCAM-1 and selectin E, is up-regulated in peripheral neuroinflammatory disease [12,13].

**Table 2 ijms-20-03022-t002:** BNB transporters.

BNB Transportes	Role in BNB	Expression in BNB-Related Disorders
Alkaline phosphatase, AP	Ionic transporter of the capillary endothelium transferring phosphate groups and preserving ionic concentrations [14].	AP has been linked to the degradation of the calcification inhibitor pyrophosphate to promote VSMC calcification [14].
Glucose transporter-1, GLUT-1	Transporter of D-glucose. It facilitates its passage into the endothelium as source of energy [15].	GLUT-1 expression in diabetic sensorimotor polyneuropathy don’t change significantly, but it is possible that diabetic condition leads to an alteration in their localization or in post-translational modification [16].
Monocarboxylate transporter 1, MCT-1	Transporter of monocarboxylic acids such as L-lactate. Under anaerobic conditions or starvation, it provides lactate as source of energy [15].	The expression levels of MCT-1 is reduced after sciatic nerve injury [17].
Creatine transporter, CRT	Transporter for creatinine that is necessary to supplies high-energy phosphate groups for the production of ATP [18].	Not so far investigated.
ABC transporters (ATP bond box), MDR-1	Efflux transporter that guarantee the outflow of xenobiotics and toxic tissue metabolism intermediates. It is fundamental for the protection of peripheral nerves from external factors [19].	Lack of MDR-1 expression leads to an increased toxicity drugs induced in BNB related disorders [20].

**Table 3 ijms-20-03022-t003:** TRLs and BNB.

Toll Like Receptors (TLRs) on BNB	Role	TLRs in BNB-Related Disorders
TLR-1	Toll like receptors are transmembrane receptors able to recognize pathogens or microbes that activate the sentinel cells of the immune system. These receptors are involved in the immune response during neurodegeneration [62,63,64].	TLR-1 is strongly induced in neurodegeneration in the sciatic nerve after injury [62].
TLR-2		TLR-2 knockout mice showed an increased rate of degenerated axons. However its absence does not influence the overall functional recovery [65].
TLR-3		TLR-3 is modestly induced in neurodegeneration in the sciatic nerve after injury [62].
TLR-4		TLR-4 is linked to neuropathic pain. In TLR-4 ko mice a decresed level of proinflammatory interleuchine 1β, interferon-γ and TNFalpha, has been showed, without a mechanical allodynia after peripheral nerve injury [66].
TRL-6		TLR-6 is modestly induced in neurodegeneration in the sciatic nerve after injury [62].
TLR-7		TLR-7 and TLR-9 are not affected in neurodegenertion in the sciatic nerve after injury [62].
TLR-9		

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
