# Peer review of "The Role of Endothelial Dysfunction in Peripheral Blood Nerve Barrier: Molecular Mechanisms and Pathophysiological Implications"

_ijms, 2019, doi:10.3390/ijms20123022_

Round 1
Reviewer 1 Report
This manuscript by Maiuolo et al. describes that the relationship between endothelial function and peripheral blood nerve barrier (BNB) in (patho)physiological states including several important diseases. Overall, it is an interesting and significant review. However, several concerns should be addressed.
Concerns
1. The manuscript, as a whole, the language and abbreviation (e.g., TNF-alpha) needs to be checked again.
2. The authors should make some tables about the roles of some key molecules as following to clearly understand them for readers.
1) Components of intracellular junctions such as claudins, occludins, and junctional adhesion molecules (Role in BNB, alterations of expressions in diseases)
2) (page 2 line 94~) Transporters (Role in BNB, alterations of expressions in diseases)
3) TLRs in BNB
3. The authors should describe detailed regulation of eNOS activities adding relationship between its activity and each phosphorylation site. In addition, how about arginase activity in BNB? Moreover, the information of NOS regulation (i.e., page 3, line 140~) were not included in Figure 1. Further, please include major sources of ROS in Figure 1a. In Figure 1B, ADMA is NOS inhibitor but not NO scavenger. Finally, the figure is too small to clearly find. Please revise Figure 1.
4. Page 5, line 171: please revise to “the soluble guanylate cyclase (sGC)”.
5. Page 5, line 176: please state more specific isozymes of PDE, like PDE5, PDE1, etc.
6. In addition of NO, other endothelium-derived factors may be also important BNB functions. Are their any evidence about prostacyclin, endothelium-derived hyperpolarizing factors, and endothelin-1?
7. Please discuss about the relationship among ER stress, NO bioavailability, and BNB function in more details. Moreover, it is better to make a figure or table about it for readers.
8. Please delete underlines in Refs [49] and [82]. And, line 666, please revise [79].
Author Response
Dear Editor and Reviewers:
Thank you very much for the valuable comments provided. We have addressed all the comments as shown in the revised manuscript. We have restructured the paragraphs, added more text, references,
redrawn and added figures as suggested by the Reviewer 1.
You will find the answer to the comments below and the added text in the manuscript in red.
Sincerely,
Jessica Maiuolo, PhD
Point by point answer
Reviewer 1
Open Review
(x) I would not like to sign my review report
( ) I would like to sign my review report
English language and style
( ) Extensive editing of English language and style required
( ) Moderate English changes required
(x) English language and style are fine/minor spell check required
( ) I don't feel qualified to judge about the English language and style
Is the work a significant contribution to the field? | |
Is the work well organized and comprehensively described? | |
Is the work scientifically sound and not misleading? | |
Are there appropriate and adequate references to related and previous work? | |
Is the English used correct and readable? |
Comments and Suggestions for Authors
This manuscript by Maiuolo et al. describes that the relationship between endothelial function and peripheral blood nerve barrier (BNB) in (patho)physiological states including several important diseases. Overall, it is an interesting and significant review. However, several concerns should be addressed.
Concerns
1. The manuscript, as a whole, the language and abbreviation (e.g., TNF-alpha) needs to be checked again.
1R. The abbreviations were checked and corrected as you suggested.
line 301 was corrected into “TNF-α”;
line 303 was corrected into “TLRs”;
line 470 was corrected from “polyphenolic bergamot fraction (BPF)” into “bergamot polyphenolic fraction (BPF)”;
the abbreviation of “inositol requiring proteins” was correct.
2. The authors should make some tables about the roles of some key molecules as following to clearly understand them for readers.
1) Components of intracellular junctions such as claudins, occludins, and junctional adhesion molecules (Role in BNB, alterations of expressions in diseases)
2) (page 2 line 94~) Transporters (Role in BNB, alterations of expressions in diseases)
3) TLRs in BNB
2R. We thank the reviewer for the suggestions. We made the tables as required: Please, see Tables 1, 2, 3.
3. The authors should describe detailed regulation of eNOS activities adding relationship between its activity and each phosphorylation site. In addition, how about arginase activity in BNB? Moreover, the information of NOS regulation (i.e., page 3, line 140~) were not included in Figure 1. Further, please include major sources of ROS in Figure 1a. In Figure 1B, ADMA is NOS inhibitor but not NO scavenger. Finally, the figure is too small to clearly find. Please revise Figure 1.
As suggested by the reviewer a detailed regulation of eNOS activity was described.
“At post-translational level, the regulation of the eNOS activity depends on the specific site of phosphorylation. Specifically the phosphorylation of eNOSSer1177 is activated in response to mechanical stimuli, such as endothelial shear stress or humoral factors. This, in turn, leads to the binding to Ca2+/calmodulin, thus increasing of eNOS activation. Then, the activity of eNOS is enhanced after Ser633 phosphorylation. This residue is located in the CaM autoinhibitory sequence of eNOS and, consequently, its phosphorylation determines eNOS activation and increased NO levels, without requiring an intracellular Ca2+ increase. Shear stress and HDL increase the phosphorylation on eNOS-Ser114 while the eNOS agonists such as VEGF, statins or bradykinin increase the phosphorylation of eNOS-Ser615. Several studies showed that the phosphorylation at this residues leads to an increased sensitivity of the enzyme to Ca2+/CaM, although the effect of the phosphorylation of this site results controversial. Further evidence shows that, the phosphorylation of eNOS at Thr495 residues leads to a reduction of activity of this enzyme; in particular, in vitro studies conducted on cultured endothelial cells demonstrated the involvement of the protein kinase C in the phosphorylation at eNOS-Thr495.”
Ref.
Mount; P.F.; Bruce, E.K.; Power,D.A. Regulation of endothelial and myocardial NO synthesis by multi-site eNOS phosphorilation. JMCC, 2007, 271-279.
You will find this part in red in the paragraph “Alterations in the production of nitric oxide at the BNB level”
“Another level of NOS regulation is related to Arginase-1 activation state. Indeed, in condition of Arginase 1 hyperactivation, there is a reduction in the amount of L-arginine, which represents the substrate of NO synthesis. Consequently, the reduction of NO levels causes an increase in ROS.”
Ref. Patel,C.; Rojas,M.; Narayanan, S.P.; et al. Arginase as mediator of diabetic retinopathy. Front. Immunol. 2013, 4,173.
You will find this part in red in the paragraph “Alterations in the production of nitric oxide at the BNB level”
The informations of NOS regulation were included as required.
Major sources of ROS were included in the new figure 2.
ADMA is now presented as a eNOS inhibitor in figure 4.
The whole fig.1 was replaced by the new figures to make all mechanisms more clear.
The required figures in the concern n.7 (i.e ER stress, NO bioavailability) has been named fig.3
4. Page 5, line 171: please revise to “the soluble guanylate cyclase (sGC)”.
4R. The sentence was corrected as you asked and is now at line 197.
5. Page 5, line 176: please state more specific isozymes of PDE, like PDE5, PDE1, etc.
5R. The sentence was corrected into “the enzyme phosphodiesterase type 5” and is now at line 202.
6. In addition of NO, other endothelium-derived factors may be also important BNB functions. Is there any evidence about prostacyclin, endothelium-derived hyperpolarizing factors, and endothelin-1?
6R. Evidences found have been added, in red, at the paragraph 4 “Endothelial dysfunction in BNB-related disorders”.
“Besides NO, endothelial cells are also the source of numerous other factors in physiological conditions or as a consequence of a vasoactive stimulus. These mediators regulate the vascular tone and the responsiveness of the endothelium by exerting different effects. The Endothelin-1 possesses inflammatory and proliferative action, inhibits eNOS, reduces NO release and antagonizes NO action. Prostacyclin and the endothelium-derived hyperpolarizing factors (EDHF) induce vasodilatation and have an antihyperproliferative effect. Several studies have also observed that EDHF may act as vasodilator in the presence of an altered bioavailability of endothelial NO.”
7. Please discuss about the relationship among ER stress, NO bioavailability, and BNB function in more details. Moreover, it is better to make a figure or table about it for readers.
7R.The relationship among ER stress, NO bioavailability and BNB was discussed. You can find it in red, at the paragraph 4 “Endothelial dysfunction in BNB-related disorders”.
“The induction of ER stress in endothelial cells involves an increased expression of endothelin-1 and a reduction of eNOS. On the contrary, it has been demonstrated that counteracting ER stress, a better activity of eNOS and a greater vascular relaxation were observed. There are several mechanisms underlying the correlation between ER stress activation and endothelial function. One of these is the insulin resistance that causes an altered production and signalling of NO. Other mechanisms include an increase in oxidative stress, due to the reduced bioavailability of NO, endothelial cell apoptosis and inflammation. In conditions of non-reversible ER stress, such as under increased ROS levels and impaired calcium homeostasis, the endothelial cell activates proapoptotic signals (i.e. JNK/p38 or caspase-12). In this context, IRE-1 stimulates JNK and P38 (MAPK) through the apoptotic signal ASK1. In turn, ASK1 inhibits eNOS by reducing phosphorylation at Ser1177 site and causes NO deficiency. On the other hand, enhanced ROS and dysregulated calcium levels directly promote apoptosis through the activation of caspase-12 (Figure 1).”
8. Please delete underlines in Refs [49] and [82]. And, line 666, please revise [79].
As reviewer suggested the underlines in refs 49 and 82 were deleted. In the new bibliography are now listed as [64] and [97]. The ref at the line 666 was corrected. In the new bibliography is now at the line 799 listed as [94]. All the bibliography has been updated and the corrections are in red.
Reviewer 2 Report
The comments:
The manuscript by Dr. Jessica Maiuolo and her collaborators presents an update in endothelial dysfunction in peripheral Blood Nerve Barrier and highlights the main mechanisms involved in the impairment of endothelial cells in specific diseases associated with peripheral nerve damage, such as diabetic neuropathy, erectile dysfunction and inflammation of the sciatic nerve.
The subject developed in this manuscript is very interesting and of potential interest for a wide readership regarding the management of patients with neurodegenerative diseases.
There are a lot of reviews and original articles studying this exciting field of Blood Nerve Barrier. However, the things are far from being clarified, especially those linked to identification of further molecular mechanisms involving in BNB disruption. For this reason, the present topic appears timely.
The manuscript is very well organized, wrote and the data are well and clear presented.
Moreover, the manuscript is state-of-the-art, comprehensive and convincing, therefore in this context my recommendation is ‘accept for publication’.
Author Response
No more action were required by Reviewer 2.